# How Large Was the Mortality Increase Directly and Indirectly Caused by the COVID-19 Epidemic? An Analysis on All-Causes Mortality Data in Italy

**DOI:** 10.3390/ijerph17103452

**Published:** 2020-05-15

**Authors:** Corrado Magnani, Danila Azzolina, Elisa Gallo, Daniela Ferrante, Dario Gregori

**Affiliations:** 1Unit of Medical Statistics and Cancer Epidemiology, Department of Translational Medicine, University of Eastern Piedmont, Novara, and CPO-Piedmont, I-28100 Novara, Italy; danila.azzolina@uniupo.it (D.A.); daniela.ferrante@uniupo.it (D.F.); 2Covid19ita Study Group, Department of Cardiac, Thoracic, Vascular Sciences, and Public Health, University of Padova, I-35131 Padova, Italy; elisa.gallo@unipd.it (E.G.); dario.gregori@unipd.it (D.G.); 3Unit of Biostatistics, Epidemiology and Public Health, Department of Cardiac, Thoracic, Vascular Sciences, and Public Health, University of Padova, I-35131 Padova, Italy

**Keywords:** SARS-CoV-2, COVID-19 epidemic, mortality, national statistics, Italy

## Abstract

*Objective:* Overall mortality is a relevant indicator of the population burden during an epidemic. It informs on both undiagnosed cases and on the effects of health system disruption. *Methods:* We aimed at evaluating the extent of the total death excess during the COVID-19 epidemic in Italy. Data from 4433 municipalities providing mortality reports until April 15th, 2020 were included for a total of 34.5 million residents from all Italian regions. Data were analyzed by region, sex and age, and compared to expected from 2015–2019. *Results:* In both genders, overall mortality was stable until February 2020 and abruptly increased from March 1st onwards. Within the municipalities studied, 77,339 deaths were observed in the period between March 1st to April 15th, 2020, in contrast to the 50,822.6 expected. The rate ratio was 1.11 before age 60 and 1.55 afterwards. Both sexes were affected. The excess was greater in the regions most affected by COVID-19 but always exceeded the deaths attributed to COVID-19. The extrapolation to the total Italian population suggests an excess of 45,033 deaths in the study period, while the number of COVID–19 deaths was 21,046. *Conclusion:* Our paper shows a large death excess during the COVID-19 epidemic in Italy; greater than the number attributed to it. Possible causes included both the undetected cases and the disruption of the Health Service organization. Timely monitoring of overall mortality based on unbiased nationwide data is an essential tool for epidemic control.

## 1. Introduction

Italy has been severely hit by coronavirus disease 2019 (COVID-19) and is currently reporting the largest outbreak in Europe together with Spain [1,2]. The first case not directly related to China was diagnosed on February 20th in a man in his 30′s living in the Lodi province (North-West Italy) [1,3]. More cases were diagnosed since then in Lombardy and the neighboring regions Veneto and Emilia-Romagna [3]. The epidemic further affected all Italian regions, with higher incidence rates in northern Italy [4]. As of April 29th, 199,470 incident cases had been diagnosed with real-time reverse transcriptase PCR (rRT-PCR) testing, with 25,215 deaths [4]. The case fatality rate was 12.6% overall and was higher in the elderly and in men [4,5].

The peak of the COVID-19 epidemic in Italy was reached in the last week of March with over 5500 new cases per day. A progressive decline has been observed since then. In the last week of April, 1552 incident cases per day have been diagnosed. A similar trend was observed for COVID-19 deaths [6].

COVID-19 epidemic containment policies were implemented, including social distancing, closure of workshops, shops and schools, and a travel ban. These policies were first implemented in the Northern regions and, since March 11th, extended to the entire country [7]. Containment policies were progressively removed from May 4th [8].

During the first phase of the epidemic, the health system was overwhelmed. Hospitals and primary care were severely affected. In particular, emergency and intensive care units (ICU) almost exclusively admitted patients affected by COVID-19 respiratory disease, while admissions for other causes and outpatient activity were reduced [9,10,11]. It is also likely that GPs reduced access to outpatient clinics and home visits, although no figures are available. Health personnel were subjected to unprecedented stress [12] and ethical dilemmas [13]. The health impact on health personnel was very important: as of April 29th, 20,831 health workers were reported as infected by COVID-19 [4]. Therefore, besides the deaths directly caused by COVID-19, it is likely that additional mortality occurred because of the health service’s restrictions.

The number of COVID-19 cases without laboratory diagnosis is debated [14]. Even the most optimistic opinions do not exclude a large number of undiagnosed cases, in particular in elderly people that present a high fatality rate. These cases would contribute additional deaths to the overall burden. Moreover, the identification of a fatality as directly and exclusively attributable to the virus can be cumbersome, particular in elderly people with multi-comorbidities [15].

Investigating the health burden of the COVID-19 epidemic requires accurate and updated figures on overall and cause-specific mortality. Davoli and colleagues issued a special report [16] based on 19 cities included in the Surveillance System of Daily Mortality [17], surveillance originally set for monitoring the effects of heatwaves. They observed from the beginning of March 2020 an increase in mortality in northern Italian cities, while little or no increase was observed in the other regions. However, the study design is not informative on mortality in smaller municipalities and is not appropriate for the estimation at the national or regional levels. The National Institute of Statistics (ISTAT) has provided a special release of mortality data for the period from January 1st, 2020, based on a mortality data rapid provision project [18]. The first report by ISS and ISTAT took into consideration mortality figures until March 31st and confirmed the large mortality increase [18]. A new data set including 4433 municipalities updated to April 15th, 2020, was also released [19].

We analyzed the mortality dataset updated to April 15th, 2020, to evaluate the pattern of changes in overall mortality until the declining phase of the epidemic. Daily all-cause mortality data from 4433 municipalities reported by the Italian National Institute of Statistics has been used from the last five years. Mortality data from January 1st to April 15th, 2020, were analyzed by region, sex, age, and compared with the expected average rates. We aimed to estimate the excess burden of deaths in addition to COVID-19-attributed mortality, and describe the time trend.

## 2. Material and Methods

We accessed the mortality data base updated to April 15th, 2020, from the ISTAT public data repository. The data included the daily number of deaths by sex and age class, for the municipalities participating in the project on fast track mortality data provision [18]. The ISTAT technical report stated that it was not a random sample but a collection of municipalities agreeing to the project, with no further selections. Data are available for the period January 1st to April 15th, 2020, and for the corresponding periods in 2015–2019 [18]. Population figures by age and sex were downloaded from the ISTAT public access repository [20]. 4433 municipalities were included, for a total of 34.5 million residents from all Italian regions (Table 1). The proportion of the population included in the sample was 57.2% overall, with a range from 20.5% (Lazio) to 88.4% (Valle D’Aosta). The study municipalities were similar to the total population of corresponding regions as regards sex ratio and proportion of subjects over 60 years.

The daily number of deaths attributed to COVID-19 and the extent of swab testing by region were obtained from the Civil Protection Department, as elaborated by the Padua University data warehouse [6]. COVID-19 deaths were defined in Italy as those occurring in patients who test positive for SARS-CoV-2 via rRT-PCR, independently from preexisting diseases that may have caused the death [5].

For the daily mortality analyses, the daily figures were first summed up by region. The daily mortality rates in 2020 were then computed dividing by the population of the study municipalities on January 1st, 2019, i.e., the most recent available population figures. The reference daily mortality rates were computed dividing the 2015–2019 average by the corresponding population. Ninety-five percent confidence intervals (95% CI) were computed assuming a Poisson distribution of the observed deaths [21]. Five-day moving averages are used to reduce random variation in the graphical presentation.

Mortality rates by region were standardized to the 2019 Italian population using the direct standardization procedure. Only all-causes mortality rates were standardized, as the age and sex structure of COVID-19 deaths could not be accessed at the regional level. Ninety-five percent confidence intervals (95% CI) of the rate ratio (RR) were computed according to Altman et al. [21].

The standardized mortality rates were compared among the regions and regressed against COVID-19 crude mortality rates using a weighed ratio estimator. The weights indicate the probability of swab testing in the Italian regions, computed as the ratio between the number of swabs and the resident population [22].

For the estimation of mortality in the total population, we multiplied the age and gender-specific mortality rates for the corresponding 2019 population, by region, under the assumption that the study municipalities were representative of the corresponding regions.

Cumulative rates were computed for the period from March 1st to April 15th summing up the daily mortality rates, by region and for the entire country.

Data were downloaded as EXCEL tables and analyzed using SAS 9.2 and R 3.6.2 [23].

Data were made available to us as frequency tables, with no information at the individual level or on the cause of death, therefore clearance from the Ethical Review Board was not necessary.

## 3. Results

In the 4433 study municipalities, 144,784 deaths were observed from January 1st, 2020, of which 77,339 occurred in the period March 1st to April 15th and are the focus of the present analyses.

Table 2 shows the number of observed deaths, the age-adjusted mortality rates (per 100,000), and the rate ratio (RR, computed dividing mortality rate in 2020 by mortality rate in 2015-2019 reference period), with 95% CIs. Given the evidence of a greater impact of COVID-19 on mortality in elderly, analyses were stratified into two age classes, using a cutoff of 60 years. Within the sample, 77,339 deaths were observed in the period between March 1st and April 15th, 2020, in contrast to an age-adjusted expected figure of 50,822.6 deaths (3417.8 under age 60 and 47,404.8 over age 60, not tabulated). In age under 60, the RR was 1.11 overall and showed a statistically significant increase only in Lombardy and in the total for Italy. In age 60 and older, the RR was 1.55 overall and showed a statistically significant increase in all regions of North Italy except Friuli-VG, in two regions of Central Italy and in the overall country.

Figure 1 presents the 5-day moving average of daily mortality rates (for 100,000 persons) in the study municipalities in 2020 and in the reference (2015–2019) years, along with COVID-19 daily mortality. Mortality was lower than in reference years until the end of February 2020. Afterwards it abruptly increased, exceeded the 2015–2019 rates on March 1st and continued increasing until the last week of March 2020. A downward change was then observed until the most recent observation (April 15th). The COVID-19 mortality rates showed a peak on April 2nd and declined since then. As for the most recent observation, the 2020 daily mortality rate was close to the sum of reference plus COVID-19 daily mortality rates. The same pattern was observed for men and women (data not in detail).

Table 3 summarizes cumulative mortality over the 45 days period from March 1st to April 15th, 2020, by region. Nationwide, the cumulative mortality rate was 222.8 deaths per every 100,000 persons, while 141.8 were expected. The excess was higher than the cumulative mortality rate due to COVID-19, which nationwide was 33.0 per every 100,000.

The increase in overall mortality was higher in the regions most affected by COVID-19. Figure 2 presents the plot of the COVID-19 mortality rates against the increase in all-causes mortality rates by region (per 100,000 inhabitants; age-adjusted), with presentation also of the frequency of swab testing. Most regions showed the same ratio of total mortality increase and COVID-19 mortality. The regression function showed slope = 0.47 (*p* < 0.001) and intercept = 7.8. Valle D’Aosta, with a large amount of swab testing activity, showed a higher COVID-19 deaths frequency compared to all-causes mortality increase.

The estimated impact of increased mortality on the total Italian population is presented in Table 4, under the assumption that the 4433 study municipalities represent an unbiased sample of the population. The overall estimated excess was 44,352.5 deaths in persons over 60 years old and 680.4 in persons younger than 60, while 21,046 deaths were attributed to COVID-19.

## 4. Discussion

The present analyses were prompted by the evidence of total mortality excess coincident with the COVID-19 epidemic in Italy [16]. Updated estimates at the regional and national levels are needed to evaluate the overall impact of the epidemic. Our research group has already built a website to report on the COVID-19 epidemic [6]. For the present work, we used available information from a public access database including 4433 municipalities with mortality data until April 15th, 2020 [19].

Our study showed a mortality increase in the period March 1st to April 15th, 2020, corresponding to the COVID-19 epidemic period. In the first 2 months of 2020, no increase over the previous five years had been observed. During the March 1st to April 15th, 2020 period, we observed a 1.55 RR over the previous five years in all-cause mortality in age class 60 and above. The RR was higher in the regions most affected by the COVID-19 epidemic, i.e., the North Italy regions and the Marche region in Central Italy. Nationwide, we estimated 45,032 excess deaths in the period. The excess was twice that explained by the COVID-19 mortality.

The peak of COVID-19 diagnoses in Italy occurred in the last week of March 2020. The highest daily increase was observed on March 21st with 6557 new cases. The highest daily number of COVID-19 deaths was observed on March 27th with 969 deaths. The peak of all-causes mortality was identified in our analyses on March 24th and anticipated the COVID-19 mortality peak by about 4 days.

This study is limited because the selection of municipalities did not follow a formal sampling design. However, the study municipalities represented over 50% of Italian municipalities, both in number and in population. Although they are not a formal sample, there are no suggestions of biased selection [18]. Preliminary analyses showed that the study municipalities were similar to the total population as regards age and sex distribution.

The mortality analyses at the regional level and the estimated excess deaths were based on age-adjusted rates, using the 2019 Italian population as standard. COVID-19 rates could not be adjusted because the distribution by age was not provided to us. The daily mortality analyses were based on crude daily rates, that are deemed sufficient given the short time span considered and the little changes in the Italian population between the reference (2015–2019) and the study (2020) years [20].

Our results are in agreement with the other reports on increased all-causes mortality associated with the COVID-19 epidemic in Italy. Davoli et al., using the Surveillance System of Daily Mortality (SSDM) [16], observed a sharp increase in total mortality from the beginning of March 2020 in the regions more severely affected by the COVID-19 epidemic. A reduction was observed after the end of March 2020, consistent with our results. This database and the SSDM had a large overlap as 13 cities were included in both. Nuvolone et al. [24] analyzed all-causes mortality in Toscana region, based on ‘ad-hoc’ collection of mortality data including 221 municipalities, for 90% of the population. They observed a 9% increased mortality in March 2020 over the previous years and estimated that COVID-19 deaths caused 63% of the excess, close to our results. Lusa applied a permutation-based approach to correct the selection bias in the analysis of a previous selection of ISTAT municipalities and identified excess mortality areas in Northern Italy [25]. The National Institute of Health (ISS) and ISTAT jointly analyzed a selection of 6866 municipalities and observed a mortality increase larger than expected from COVID-19 figures [18]. That data base was larger but was updated only to March 31st and therefore is not informative about the recent evolution of the COVID-19 epidemic.

The increase of overall or unrelated mortality rates has been used in several situations as an epidemiological signal, such as for tuberculosis and AIDS [26,27], influenza [28], chikungunya [29,30], and for general surveillance [31]. In the present case, the epidemic was so important that the direct observation of COVID-19 cases showed enough evidence for the recognition [1]. However, increased all-cause mortality provides important signals on the global health impact.

Our analyses indicate that the all-causes mortality trend anticipated the increase of COVID-19 mortality and also showed a more rapid reduction. Possible causes of the observed increase include the large size of undiagnosed severe COVID-19 cases, the reduced access to health services due to the disruption of the normal work processes, the forced reassurance of sick patients affected by other diseases who avoided accessing hospitals or clinics for fear of contamination, and possibly other mechanisms. The effect of the COVID-19 epidemic on the Italian health system in terms of ICU admissions and hospitalizations has been analyzed in detail elsewhere [32] and results are only summarized here. ICU admissions showed a sharp increase from the last days of February until reaching a peak of 4208 admissions on April 3rd, followed by a decreasing trend (Appendix A). The observed reduction in the all-causes mortality trend could correspond to the return of the Health System to normal conditions but could also indicate a harvesting effect if the previous increase was determined by the anticipation of deaths.

The containment policies have been demonstrated to be effective for reducing the COVID-19 diffusion rate [32]. However, the number of ICU admissions is also related to the preventive actions implemented by the health care decision-makers. For example, a broad testing strategy, as adopted by the Veneto region, was associated with a reduction in the ICU admission rate [33]. An eventual mitigation of the mortality increase is therefore expected. From a public health perspective, what matters is identifying modifiable factors that increase the risk of severe disease or death, regardless of the direct microbial involvement [34].

Disentangling the different hypotheses requires extended observation and the in-depth investigation of individual cases. In particular, future observation should focus on the distinction between transient-harvesting-phenomenon or actual, cause-specific, increased mortality. This study is limited in this respect because of the absence of information on the causes of death, which precludes finer interpretation of the results with respect to the different hypotheses of additional COVID-19 related deaths, additional unrelated deaths or harvesting. It is mandatory to continue the provision of ‘fast track’ data, with unbiased sampling design, to analyze the possible causes and mechanisms. In any case, our results highlight the need for timely and unbiased mortality data as an essential tool for public health.

## 5. Conclusions

Our analyses showed a large increase in all-causes mortality during the COVID-19 epidemic in Italy. The excess was greater than the COVID-19 related mortality. Possible explanations include both undiagnosed COVID-19 cases and the limitations in access to health services.

## Figures and Tables

**Figure 1 ijerph-17-03452-f001:**
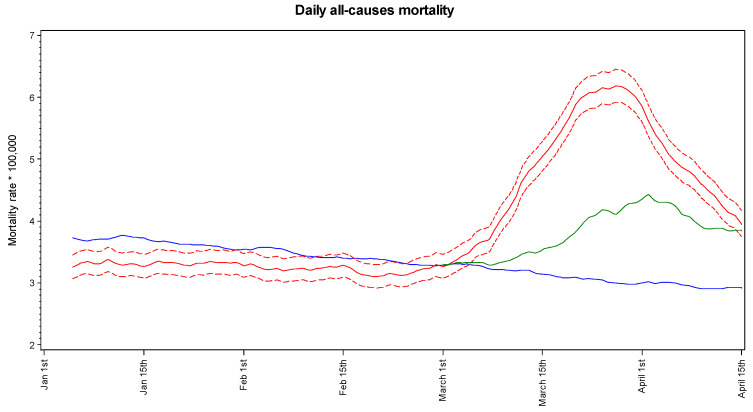
Daily mortality rates (5-day moving average) in the total sample population (red lines: mean and 95% confidence interval) and in the average of 2015–2019 periods (blue line), and COVID-19 mortality rates (green line).

**Figure 2 ijerph-17-03452-f002:**
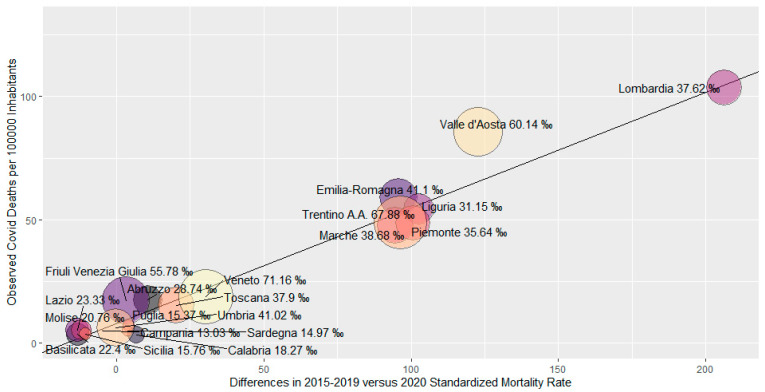
Differences between the 2015–2019 and 2020 Standardized Mortality Rates versus COVID mortality rates per 100,000 inhabitants calculated via weighed ratio estimator. The weights correspond to the probability of swab testing among the population for Italian regions (as of April 30^th^). The size of the circle is proportional to the number of swabs over 1000 inhabitants. The label of the circle includes the name of the region and the number of swabs over 1000 inhabitants. The estimated linear regression line is reported (intercept *t* = 7.8, slope = 0.47, *p* value < 0.001).

**Table 1 ijerph-17-03452-t001:** Descriptive data on the study municipalities and the corresponding regions.

Region	Area	Study Municipalities				Total Region	
		Total Population	Age ≥ 60	M/F	% Of Region Population	Age ≥ 60	M/F
Piemonte	North	2,600,917	32.2%	0.95	59.7%	32.2%	0.94
Valle d’Aosta	North	111,131	30.7%	0.95	88.4%	30.4%	0.96
Lombardia	North	7,879,002	28.7%	0.96	78.3%	28.7%	0.96
Trentino-AA	North	421,612	28.6%	0.95	39.3%	26.8%	0.97
Veneto	North	3,554,750	29.3%	0.96	72.5%	29.2%	0.96
Friuli-VG	North	431,722	31.8%	0.97	35.5%	32.7%	0.95
Liguria	North	1,190,066	35.0%	0.91	76.7%	35.3%	0.92
Emilia-R.	North	3,738,441	30.3%	0.94	83.8%	30.3%	0.95
Toscana	Centre	2,978,743	31.7%	0.93	79.9%	31.8%	0.93
Umbria	Centre	632,162	31.4%	0.92	71.7%	32.0%	0.93
Marche	Centre	730,691	31.5%	0.94	47.9%	31.3%	0.94
Lazio	Centre	1,208,119	28.0%	0.96	20.5%	28.0%	0.93
Abruzzo	South	740,612	30.5%	0.96	56.5%	30.5%	0.95
Molise	South	169,028	30.9%	0.98	55.3%	31.5%	0.97
Campania	South	1,503,111	25.7%	0.95	25.9%	25.0%	0.95
Puglia	South	2,427,922	28.9%	0.94	60.3%	28.4%	0.95
Basilicata	South	128,087	29.6%	0.97	22.8%	29.8%	0.97
Calabria	South	686,627	27.7%	0.97	35.3%	28.1%	0.96
Sicilia	South	2,499,214	27.7%	0.94	50.0%	27.6%	0.95
Sardegna	South	904,997	30.1%	0.98	55.2%	30.9%	0.97
Italy		34,536,954	29.7%	0.95	57.2%	29.2%	0.95

**Table 2 ijerph-17-03452-t002:** Observed (Obs.) deaths, age-adjusted mortality rate per 100,000 (Mortality rate), Rate Ratio (RR) and 95% CIs in the study period from March 1st to April 15th, 2020, in the study municipalities, by region and age class (< 60 vs.≥ 60).

	Area	Age < 60				Age ≥ 60			
		Obs. Deaths	Mortality Rate	RR	95% CI	Obs. Deaths	Mortality Rate	RR	95% CI
Piemonte	North	279	15.7	1.02	(0.71–1.33)	6675	801.0	1.64	(1.53–1.74)
Valle d’Aosta	North	7	9.1	0.72	(0.00–1.61)	311	907.4	1.83	(1.28–2.38)
Lombardia	North	1176	21.0	1.63	(1.39–1.87)	25834	1130.7	2.61	(2.53–2.70)
Trentino-AA	North	41	13.3	1.09	(0.29–1.90)	969	783.2	1.83	(1.52–2.14)
Veneto	North	319	12.7	0.97	(0.70–1.25)	5682	544.0	1.24	(1.15–1.33)
Friuli-VG	North	38	13.0	0.9	(0.22–1.58)	648	496.0	1.03	(0.81–1.24)
Liguria	North	120	15.6	0.98	(0.58–1.39)	3315	787.8	1.59	(1.44–1.73)
Emilia-R.	North	448	17.2	1.33	(1.00–1.66)	8843	779.4	1.65	(1.56–1.74)
Toscana	Centre	262	12.9	0.99	(0.68–1.30)	5015	532.1	1.14	(1.05–1.22)
Umbria	Centre	59	13.7	1.07	(0.39–1.75)	922	467.1	0.99	(0.82–1.17)
Marche	Centre	86	17.1	1.38	(0.62–2.15)	1730	755.6	1.63	(1.42–1.84)
Lazio	Centre	89	10.4	0.66	(0.31–1.00)	1431	440.2	0.93	(0.80–1.06)
Abruzzo	South	73	14.2	0.94	(0.41–1.47)	1173	523.1	1.08	(0.91–1.24)
Molise	South	19	16.3	0.97	(0.00–2.02)	254	496.1	0.93	(0.62–1.24)
Campania	South	167	14.8	0.87	(0.53–1.21)	1795	454.1	0.98	(0.85–1.10)
Puglia	South	251	14.4	1.03	(0.68–1.39)	3201	454.5	1.03	(0.93–1.13)
Basilicata	South	3	3.3	0.23	(0.00–0.59)	189	478.1	0.96	(0.59–1.33)
Calabria	South	62	12.7	0.78	(0.26–1.30)	987	526.3	1.07	(0.89–1.25)
Sicilia	South	216	12.0	0.74	(0.46–1.01)	3348	486.2	0.95	(0.86–1.04)
Sardegna	South	94	15.0	0.85	(0.41–1.28)	1208	452.4	1.04	(0.88–1.20)
Italy		3809	15.6	1.11	(1.02–1.21)	73530	708.1	1.55	(1.52–1.58)

**Table 3 ijerph-17-03452-t003:** Cumulative mortality (per 100,000) from March 1st to April 15th in the study municipalities, overall and attributed to COVID-19, by region.

	Area	Cumulative Mortality (Per 100.000)			
		Expected (2015–2019)	Study Period (2020)	Difference	COVID
Piemonte	North	160.5	264.3	103.8	39.7
Valle d’Aosta	North	154.1	277.9	123.8	89.0
Lombardia	North	129.2	339.9	210.7	106.1
Trentino-AA	North	128.3	235.9	107.6	46.3
Veneto	North	133.2	167.6	34.4	17.4
Friuli-VG	North	150.1	161.7	11.6	15.9
Liguria	North	180.8	285.6	104.8	48.3
Emilia-R.	North	149.0	246.4	97.4	57.2
Toscana	Centre	152.6	176.1	23.5	13.3
Umbria	Centre	150.6	156.2	5.6	5.9
Marche	Centre	149.3	246.8	97.5	46.1
Lazio	Centre	132.5	127.7	−4.8	4.8
Abruzzo	South	153.5	169.9	16.4	16.3
Molise	South	166.3	163.1	−3.2	4.7
Campania	South	130.6	131.2	0.6	4.2
Puglia	South	131.0	142.4	11.4	6.4
Basilicata	South	156.8	151.9	−4.9	3.1
Calabria	South	140.8	152.4	11.6	3.4
Sicilia	South	147.3	143.9	−3.4	3.2
Sardegna	South	132.1	147.1	15.0	4.5
Italy		141.8	222.8	81.0	33.0

**Table 4 ijerph-17-03452-t004:** Estimated excess deaths from March 1st to April 15th, 2020 in the total population, by region.

	Area	Estimated Excess Deaths in the Total Population		Deaths Attributed to COVID
		Age < 60	Age ≥ 60	
**Piemonte**	**North**	9.6	4371.5	1927
Valle d’Aosta	North	−3.1	157.4	118
Lombardia	North	581.7	20,150.8	11,125
Trentino-AA	North	9.1	1024.3	524
Veneto	North	−11.8	1506.2	904
Friuli-VG	North	−11.4	53.2	206
Liguria	North	−2.6	1593.2	793
Emilia-R.	North	132.6	4138.9	2703
Toscana	Centre	−2.9	762.0	538
Umbria	Centre	5.6	−7.6	53
Marche	Centre	49.6	1397.1	728
Lazio	Centre	−231.6	−543.0	300
Abruzzo	South	−8.3	146.6	232
Molise	South	−1.0	−34.8	15
Campania	South	−95.2	−168.3	260
Puglia	South	13.5	152.2	278
Basilicata	South	−43.9	−30.6	19
Calabria	South	−50.4	181.0	68
Sicilia	South	−154.1	−366.0	175
Sardegna	South	−30.4	94.0	80
Italy		680.4	44,352.5	21,046

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
