# Peer review of "How Large Was the Mortality Increase Directly and Indirectly Caused by the COVID-19 Epidemic? An Analysis on All-Causes Mortality Data in Italy"

_ijerph, 2020, doi:10.3390/ijerph17103452_

Round 1

Reviewer 1 Report

The authors are presenting results on the analysis of overall mortality in Italy in the course of the COVID-19 outbreak. The mortality excess is computed from reported numbers of a large Italian sample of municipalties and expected average daily rates computed from 2015 to 2019. The estimated mortality excess has been compared with the actual reported number of deaths in COVID-19 patients. Key message of the paper is the large discrepancy of officially reported COVID-19-related mortality (4825) to the estimated maximal mortality excess (39867) within the first three weeks of 2020. The large difference is related to the collapse of the health care system (with a large number of additional deaths caused by excessive demand) and an unsatisfactory viral testing in this crisis. Therefore, the authors conclude that a timely monitoring of the overall mortality constitutes a better reporting tool in the event of a pandemic to measure the impact of the crisis and to lower the reporting bias.

Personal impression:

First, I went through the article quickly,to get a brief overview and my impression was not that exciting. I had the feeling that the article doesn’t contain any new information and the illustrations made me thinking, that the authors just want to show that the mortality rates in Italy are higher than usual. That’s actually not surprising I thought.

Only after reading the article in detail, my impression changed and I really got the impact of the paper: there are a lot more people deceasing during the crisis than we would expect, even if you add the official deadly cases with COVID-19 to the average expected deaths. The large mortality excess is probably caused by estimated numbers of unknown cases and far more people died from other illnesses during the crisis caused by the overburdening of healthcare system.

Unfortunately, the authors have not written the article that concisely and in an intelligible fashion. The English absolutely needs to be improved. But I see great chances to improve the paper significantly overall.

Please provide the following information within the manuscript:

1. Were all deceased people tested for COVID-19 disease? If yes, since when?
This is important that the ready can judge the unreported number of deadly cases.

2. Please provide information about the overburdening of the health care system. When was the lock-down starting? Are there dates reported for the overcrowding of ICU capacities?

3. There are regions/municipalities in Italy (especially in the south) with very low numbers of infections in the first three weeks of March. Was the all-cause mortality also above average (due to nation-wide lock-down)? Please point out possible reasons for such a development in the discussion.

Line-by-line comments and questions on the manuscript:

15-28: Please keep sentences short and clear. Focus on the results of the study. Work on grammar and style.
24: Isn’t it just an extrapolation/projection to a full sample rather than an upper limit?
32: better: by analyzing excess mortality
33: better: all-cause mortality
32-36:
better: In this a fast-track project, daily all-cause mortality data from 1084 municipalities reported by the Italian National Institute of Statistics has been used from the last five years. Mortality data from 1 January to 21 March 2020 were analyzed by region, sex, age, and compared with the expected average rates.
36: What do you mean with ‘The extent of swab testing was considered’? Do you mean ‘Mortality excess was correlated with swab testing for COVID-19’?
36: I don’t see simulations in this study. Probably you mean something different?
38: better: Within the sample, 16216 deaths were observed in the period between 1
st and 21st March 2020 in contrast to an expected five-year average of 7843 deaths.
40: better: The discrepancy was more present in regions affected by COVID-19.
40: age classes ‘61-65’ and ‘71-75’ are not used in the main text. So please remove th
ese age classes from the abstract. Also, significant increases are also present in the young age groups <=60 (see Table 1).
4
3: I recommend to put this sentence in the methods: ‘We aim to analyze total death excess during the first three weeks of March in the COVID-19 epidemic in Italy.’
47: I don’t see, that this is an upper limit.
53: ‘interested’?
58: abbreviation here, instead in line 91: corona virus disease 2019 (COVID-19)
61: health system
61ff: wording. e.g. ‘resources’,’dedicated’
67: better: Several MDs got infected because …
69: better: deaths directly caused by SARS-CoV-2 infections
72: Make two sentences out of it.

Introduction: Are there any other papers to refer to for such an analysis of the mortality excess in pandemic situations?

120: The daily number of deaths in each municipality will change with the number of inhabitants. Have you accounted for that while computing the average? This would be in line with the weighed ratio estimator described in line 129ff.
123: Why do you want to calculate an upper limit for entire Italy? It would be more precise and more meaningful if you just compare the reported COVID-19 deaths (C) and all-cause-mortalty (A) in the regions from which you have data. From the last five years you compute the expect normal mortality (E) and the discrepancy D = A-E-C is the number of deaths which is expected to contain the undetected cases and deaths directly or indirectly caused by the collapse of the health care system and the lock-down effects. Since this number D is more than C, the message could be that we all should have a much closer look to the all-cause mortality rate rather than the daily reported COVID-19 deaths to get an impression of the real impact of the pandemic.
126: keep the notation as before: January 1st 2020
126: Do you have accounted for different age distributions in the sample and the corresponding regional age distributions to estimate the number of total deaths in each region?

Table 1: Commas needs to be replace by dots.
Header: remove ‘(sum of M&f)’, last column: remove ‘in each area’
Probably the table would be
more readable if the observed deaths and expected deaths are reported side-by-side in a single column with the percent change in brackets for each age group: e.g., Piemonte: 42 | 39.8 (+5.5%) and 992 | 556.8 (+78.2%) or in this way: +5.5% (42 vs. 39.8). The reader would directly see the connection of your measures.
I don’t get, what the last two columns are representing. Is this the number of death not related to the COVID-19 or the normal death rate?

159: So far the sub-figures of men and women are not left and right. I don’t see a huge difference between men and women, so I suggest to present the overall numbers in one figure and suggest to place a sex-specific figure as a supplement.

Figure 1: Would it make sense to plot the moving average (7 days length) instead of daily numbers to get rid of heavy fluctuations?

The axis should be changed to dates instead of days after new years eve. Remove the sub-header ‘cod_reg=Italia’. Please add a third line which is the expected number of deaths plus the number of COVID-19 deaths in the sample regions, to get an impression of the discrepancy. Set the upper limit of the x-axis to the 21st March.

Figure 2: Please use the ggrepel for non-overlapping labels: http://www.sthda.com/english/wiki/ggplot2-texts-add-text-annotations-to-a-graph-in-r-software
It would be nice to add a diagonal line
(x=y) to directly compare the the total death increase with the number of COVID-19 deaths. This would better capture the essence of the figure, providing the fact that the total death increase cannot only be explained by the COVID-19 deaths. This affects all regions but Valle d’Aosta.
Instead of bubbles, can you also provide the swab testing rate for each region, e.g, as another label?

178: Hereafter I would conclude, the difference between 16216 and expected 7843.4 deaths is about 8400. Although 4825 COVID-19 deaths were reported for entire Italy and taken into account those as additional death cases, there is still an unexplained mortality excess of about 3600 deaths.
184: I really don’t think that the maximal excess is here of major interest. With the observed data from your sample and the reported number of nation-wide COVID-19 deaths you rather are able to provide a minimum number of excess (assuming similar daily mortality rates for regions were you don’t have data from), which is as already stated to be 3600 additional deaths (mainly as a side-effect of the overburdening of the health care system).
225: remove additional space
227: COVID-19

Author Response

Reviewer 1:

  1. Were all deceased people tested for COVID-19 disease? If yes, since when?
    This is important that the ready can judge the unreported number of deadly cases.

Answer:

Those classified as COVID-19 deaths were tested, as the classification as COVID-19 case requires a positive test (see ref 6). No information are available on decedents with negative test.

The numbers of COVID-19 cases and of COVID-19 deaths, as reported in the introduction (lines 39 – 45), in figure 1 and elsewhere are the number of subjects officially classified as caused by COVID, according to PCR testing and to clinical evaluation. The number is provided daily by the Ministry of Health / Department of Civil Protection (ref 5).

To provide a better information we changed the text regarding certified COVID deaths in MM (line 98):

COVID-19–related deaths were defined in Italy as those occurring in patients who test positive for SARS-CoV-2 via RT-PCR, independently from preexisting diseases that may have caused death [Onder].

  1. Please provide information about the overburdening of the health care system. When was the lock-down starting? Are there dates reported for the overcrowding of ICU capacities?

Answer:

Further details were provided in the introduction (lines 42 – 60). This section has been entirely rewritten.

  1. There are regions/municipalities in Italy (especially in the south) with very low numbers of infections in the first three weeks of March. Was the all-cause mortality also above average (due to nation-wide lock-down)? Please point out possible reasons for such a development in the discussion.

Answer:

Data were provided by region, both in the tables and in figure 2.  It is relevant that those regions in southern Italy are reputed the worst as to the quality of health system but nevertheless did not show death increses. However we prefer to avoid too strong comments on the quality of health system in the different regions as they exceed the purpose of the paper.

Line-by-line comments and questions on the manuscript:

The text has been largely rewritten and it was impossible to track the changes. However all the comments by the reviewer have been addressed.

15-28: Please keep sentences short and clear. Focus on the results of the study. Work on grammar and style.

Answer: done with the help of an English mother language collaborator.

24: Isn’t it just an extrapolation/projection to a full sample rather than an upper limit?

Answer: changed and made clearer. It is an extrapolation

32: better: by analyzing excess mortality

Answer: changed

33: better: all-cause mortality

Answer: changed

32-36: better: In this a fast-track project, daily all-cause mortality data from 1084 municipalities reported by the Italian National Institute of Statistics has been used from the last five years. Mortality data from 1 January to 21 March 2020 were analyzed by region, sex, age, and compared with the expected average rates.

Answer: changed in both abstract and text (Abstract; lines 78 and following)

36: What do you mean with ‘The extent of swab testing was considered’? Do you mean ‘Mortality excess was correlated with swab testing for COVID-19’?

Answer: statement was erased from the abstract and modified in the text (lines 113-116)

36: I don’t see simulations in this study. Probably you mean something different?

Answer: It was a misspelling. Only extrapolations were done.

38: better: Within the sample, 16216 deaths were observed in the period between 1st and 21st March 2020 in contrast to an expected five-year average of 7843 deaths.

Answer: changed as suggested. (Abstract and lines 132-134)

40: better: The discrepancy was more present in regions affected by COVID-19.

Answer: text was reworded

40: age classes ‘61-65’ and ‘71-75’ are not used in the main text. So please remove these age classes from the abstract. Also, significant increases are also present in the young age groups <=60 (see Table 1).

Answer: text was reworded as suggested.

43: I recommend to put this sentence in the methods: ‘We aim to analyze total death excess during the first three weeks of March in the COVID-19 epidemic in Italy.’

Answer: text was reworded as suggested (line 16).

47: I don’t see, that this is an upper limit.

Answer: text was reworded

53: ‘interested’?

Answer: The statement was changed to “Since the last days of February 2020 Italy is hit by the SARS-CoV-2 [1 , 2] epidemic. (lines 33-34)

58: abbreviation here, instead in line 91: corona virus disease 2019 (COVID-19) 

Answer:  Changed as suggested (line 33)

61: health system

Answer:   Changed as suggested

61ff: wording. e.g. ‘resources’,’dedicated’

Answer: reworded (line 53)

67: better: Several MDs got infected because …

Answer:   Changed as suggested

69: better: deaths directly caused by SARS-CoV-2 infections

Answer:   Changed as suggested

72: Make two sentences out of it.

Answer:   Changed as suggested

Introduction: Are there any other papers to refer to for such an analysis of the mortality excess in pandemic situations?

Answer: There are several examples, that we included in the discussion (refs 26 – 31)

120: The daily number of deaths in each municipality will change with the number of inhabitants. Have you accounted for that while computing the average? This would be in line with the weighed ratio estimator described in line 129ff.

Answer: In the new analyses we focused on rates.

123: Why do you want to calculate an upper limit for entire Italy? It would be more precise and more meaningful if you just compare the reported COVID-19 deaths (C) and all-cause-mortalty (A) in the regions from which you have data. From the last five years you compute the expect normal mortality (E) and the discrepancy D = A-E-C is the number of deaths which is expected to contain the undetected cases and deaths directly or indirectly caused by the collapse of the health care system and the lock-down effects. Since this number D is more than C, the message could be that we all should have a much closer look to the all-cause mortality rate rather than the daily reported COVID-19 deaths to get an impression of the real impact of the pandemic.

Answer: The new analyses include the estimation of excess deaths in the total Italian population, extrapolating from the study municipalities that include about 50% of Italian population.

126: keep the notation as before: January 1st 2020

Answer:   Changed as suggested

126: Do you have accounted for different age distributions in the sample and the corresponding regional age distributions to estimate the number of total deaths in each region?

Answer: In the new analyses we focused on rates, adjusted by age. The topic is also discussed in lines 208 – 213.

Table 1: Commas needs to be replace by dots.

Answer:   Changed as suggested

Header: remove ‘(sum of M&f)’, last column: remove ‘in each area’ 

Answer:   Changed as suggested

Probably the table would be more readable if the observed deaths and expected deaths are reported side-by-side in a single column with the percent change in brackets for each age group: e.g., Piemonte: 42 | 39.8 (+5.5%) and 992 | 556.8 (+78.2%) or in this way: +5.5% (42 vs. 39.8). The reader would directly see the connection of your measures.
I don’t get, what the last two columns are representing. Is this the number of death not related to the COVID-19 or the normal death rate?

Answer: the table was changed

159: So far the sub-figures of men and women are not left and right. I don’t see a huge difference between men and women, so I suggest to present the overall numbers in one figure and suggest to place a sex-specific figure as a supplement.

Answer: the table was changed

Figure 1: Would it make sense to plot the moving average (7 days length) instead of daily numbers to get rid of heavy fluctuations?

Answer: 5 days moving averages were used (figure 1).

The axis should be changed to dates instead of days after new years eve. Remove the sub-header ‘cod_reg=Italia’. Please add a third line which is the expected number of deaths plus the number of COVID-19 deaths in the sample regions, to get an impression of the discrepancy. Set the upper limit of the x-axis to the 21st March.

Answer: changed as suggested.

Figure 2: Please use the ggrepel for non-overlapping labels: http://www.sthda.com/english/wiki/ggplot2-texts-add-text-annotations-to-a-graph-in-r-software
It would be nice to add a diagonal line (x=y) to directly compare the the total death increase with the number of COVID-19 deaths. This would better capture the essence of the figure, providing the fact that the total death increase cannot only be explained by the COVID-19 deaths. This affects all regions but Valle d’Aosta.
Instead of bubbles, can you also provide the swab testing rate for each region, e.g, as another label?

Answer: changed as suggested.

178: Hereafter I would conclude, the difference between 16216 and expected 7843.4 deaths is about 8400. Although 4825 COVID-19 deaths were reported for entire Italy and taken into account those as additional death cases, there is still an unexplained mortality excess of about 3600 deaths.

Answer: The reviewer comment is correct and we changed the text to make it cleare to the reader (lines 120 – 145).

184: I really don’t think that the maximal excess is here of major interest. With the observed data from your sample and the reported number of nation-wide COVID-19 deaths you rather are able to provide a minimum number of excess (assuming similar daily mortality rates for regions were you don’t have data from), which is as already stated to be 3600 additional deaths (mainly as a side-effect of the overburdening of the health care system).

Answer: the text was reworded

225: remove additional space

Answer: Changed as suggested

227: COVID-19 

Answer: No, it was not an error. However the text was changed.

Reviewer 2 Report

The paper by Magnani et al. aims to quantify the excess burden of deaths in Italy over the period March 1-21, 2020. The authors had access to the official mortality data of 1084 out of 7904 municipalities, and estimated the national number of deaths directly or indirectly associated with COVID-19 by projecting a 20%-increase in mortality on non-sampled populations. A major limitation is that the observed number of deaths was obtained via nonprobability sampling, but this is largely acknowledged by the authors. I think that the work has overall merit, but the study aim is not well substantiated, and parts of the results are equivocal. Some English changes are also necessary. Here are my comments:

1) Line 53: Please use the present perfect (“[…] has been interested […]”).

2) Line 62: There is a repetition.

3) Lines 90-3: The added value of this work as compared to Davoli’s is that the authors tried to quantify excess mortality in all the municipalities of Italy with a sort of population projection (this is explained pretty well in lines 122-8). This is not clear though when I read the aim of the study. The authors say that they “extended” the monitoring activity of excess mortality—I think they should elaborate more on what they mean by “extension”.

4) Line 97: “Information” asks for singular verb forms.

5) Lines 129-34: I am a bit confused, since the authors say here that they illustrated the observed increase in sampled municipalities, while it seems to me that they showed the increase projected on the entire populations of each region. What is the right interpretation of the x-axis in Figure 2?

6) Table 1: I think the authors summarized deaths from March 1 to March 21, but this is not mentioned in the table caption.

7) Lines 159-60: The two panels of Figure 1 are one above the other, not one next to each other.

8) Figure 1: Please erase “cod_reg=Italia” and update x-axis labels by showing dates instead of days from January 1.

9) Figure 2: I think that presenting the extent of swab testing through bubble sizes is unnecessary and misleading at the same time. Unnecessary because there is a strong linear relationship between the number of tests and the number of COVID-19 deaths, and this is no surprise to me (the more cases are detected, the more deaths are registered); misleading because one would be tempted to draw disproportionate conclusions by looking at this graph. Let us look for example at Liguria and Marche, two regions that have the same COVID-19 mortality but different increases in all-cause mortality: in the Marches there might have been a larger proportion of undetected COVID cases, or maybe the hospitals were really overwhelmed and people could not be treated for other conditions. To date we are not able to answer these questions and I bet that the authors know that, because they do not discuss or provide any interpretation for this figure. Another problem is that the outlying regions are those with small population sizes (Aosta Valley, Molise, Basilicata…), and their estimates are much less precise than those of the other regions. Non-Italian readers do not know that. In conclusion, I suggest deleting this graph.

10) General comment: Please be consistent when you name Italian regions (e.g. Valle d’Aosta versus Aosta Valley).

Author Response

The text has been largely rewritten and it was impossible to track the changes. However all the comments by the reviewer have been addressed.

1) Line 53: Please use the present perfect (“[…] has been interested […]”).

Answer: changed as suggested

2) Line 62: There is a repetition.

Answer: changed as suggested

3) Lines 90-3: The added value of this work as compared to Davoli’s is that the authors tried to quantify excess mortality in all the municipalities of Italy with a sort of population projection (this is explained pretty well in lines 122-8). This is not clear though when I read the aim of the study. The authors say that they “extended” the monitoring activity of excess mortality—I think they should elaborate more on what they mean by “extension”.

Answer: the text was changed as suggested (lines 77 – 82)

4) Line 97: “Information” asks for singular verb forms.

Answer: changed as suggested

5) Lines 129-34: I am a bit confused, since the authors say here that they illustrated the observed increase in sampled municipalities, while it seems to me that they showed the increase projected on the entire populations of each region. What is the right interpretation of the x-axis in Figure 2?

Answer: the description was made clearer (lines 165 – 170 and caption of figure 2)

6) Table 1: I think the authors summarized deaths from March 1 to March 21, but this is not mentioned in the table caption.

Answer: changed as suggested

7) Lines 159-60: The two panels of Figure 1 are one above the other, not one next to each other. 

Answer: changed  presenting one panel only for total mortality. Results by sex are limited to the text and tables.

8) Figure 1: Please erase “cod_reg=Italia” and update x-axis labels by showing dates instead of days from January 1.

Answer: changed as suggested

9) Figure 2: I think that presenting the extent of swab testing through bubble sizes is unnecessary and misleading at the same time. Unnecessary because there is a strong linear relationship between the number of tests and the number of COVID-19 deaths, and this is no surprise to me (the more cases are detected, the more deaths are registered); misleading because one would be tempted to draw disproportionate conclusions by looking at this graph. Let us look for example at Liguria and Marche, two regions that have the same COVID-19 mortality but different increases in all-cause mortality: in the Marches there might have been a larger proportion of undetected COVID cases, or maybe the hospitals were really overwhelmed and people could not be treated for other conditions. To date we are not able to answer these questions and I bet that the authors know that, because they do not discuss or provide any interpretation for this figure. Another problem is that the outlying regions are those with small population sizes (Aosta Valley, Molise, Basilicata…), and their estimates are much less precise than those of the other regions. Non-Italian readers do not know that. In conclusion, I suggest deleting this graph.

Answer: Figure 2 was changed. Swab testing was a relevant activity for the management of the epidemic and we are reluctant to cancel it. Moreover no such an indication was provided by the other reviewers.

10) General comment: Please be consistent when you name Italian regions (e.g. Valle d’Aosta versus Aosta Valley).

Answer: changed as suggested

Reviewer 3 Report

The mortality in patients with Covid-19 is very important. It’s currently a “hot topic”, and the authors has made significant contributions. However, this study has several limitations. Sampling for mortality is not random, therefore, this probably leads to bias. There is also probably a registry of deaths from other causes non-COVID-19.

Despite this, the author could strengthen the paper through the following:

Methods:
The estimates are correct, but they need to update the information. Italy has more than 180 thousand cases and more than 25 thousand deaths by COVID-19 (to date).

Resultados:
Include a Figure on absolute total daily mortality in the overall sample data (to date). I also suggest including an age-adjusted estimate of mortality in a Figure (to date).

Discussion
Include a paragraph on the conclusion limited to your findings.

Author Response

The mortality in patients with Covid-19 is very important. It’s currently a “hot topic”, and the authors has made significant contributions. However, this study has several limitations. Sampling for mortality is not random, therefore, this probably leads to bias. There is also probably a registry of deaths from other causes non-COVID-19.

Answer: A new data set was made available, not affected by selection bias. All the analyses and data presentation have been updated accordingly.

Despite this, the author could strengthen the paper through the following:

Methods:
The estimates are correct, but they need to update the information. Italy has more than 180 thousand cases and more than 25 thousand deaths by COVID-19 (to date).

Answer: We reported the figures at the time of preparation of the paper. These figures have been updated as new data were analyzed.

Resultados:
Include a Figure on absolute total daily mortality in the overall sample data (to date). I also suggest including an age-adjusted estimate of mortality in a Figure (to date).

Answer: The information was provided.

Discussion
Include a paragraph on the conclusion limited to your findings.

Answer: The paragraph was added

Round 2

Reviewer 1 Report

The authors made substantial changes of the entire manuscript and updated their analysis by taking data from a more recent time window. The authors presented their results in an intelligible fashion and the paper is now well written.

Some minor changes:

44: bracket missing '[7]'
72: superscript 'th'
75: January 1st to April 15th
85: 4433
86: 34.5 million
88: sex ratio
88: remove the underlining
88: 'similar' among the 5 years?
100: Five-day
105: small capitals: rate ratio
109: space after population
123: are these 'age-adjusted daily mortality rates (per 100,000)'?
136: 5-day

Author Response

We would like to thank the reviewer for the constructive comments.

All the requested modifications were done.

44: bracket missing '[7]'
72: superscript 'th'
75: January 1st to April 15th
85: 4433
86: 34.5 million
88: sex ratio
88: remove the underlining

100: Five-day
105: small capitals: rate ratio
109: space after population

136: 5-day

88: 'similar' among the 5 years? The phrase was changed as follows: The study municipalities were similar to the total population of corresponding regions as regards sex ratio and proportion of subjects over 60 years.

123: are these 'age-adjusted daily mortality rates (per 100,000)'? Yes, they are. Both the text and the table caption were changed.

Reviewer 2 Report

The paper has improved. Well done.

Author Response

Thank-you.

Reviewer 3 Report

Include and discuss the limitations of your study in Discussion section.

Author Response

We thank the reviewer for the constructive comments. 

Two statements were added about the study limitations. The new paragraphs of interest are as follows:

Lines 201 – 205: This study is limited because the selection of municipalities did not follow a formal sampling design. However, the study municipalities represented over 50% of Italian municipalities, both in number and in population. Albeit they are not a formal sample there are no suggestions of biased selection [18]. Preliminary analyses showed that the study municipalities were similar to the total population as regards age and sex distribution.

251-254: This study in limited in this respect because of the absence of information on the causes of death, that precludes finer interpretation of the results in respect of the different hypotheses of additional COVID-19 related deaths, additional unrelated deaths or harvesting.